# Ecological Niche Differentiation and Distribution Dynamics Revealing Climate Change Responses in the Chinese Genus *Dysosma*

**DOI:** 10.3390/plants15010162

**Published:** 2026-01-05

**Authors:** Rui Chen, Fangming Luo, Weihao Yao, Runmei Yang, Lang Huang, He Li, Mao Li

**Affiliations:** 1Guizhou Academy of Forestry, Guiyang 550005, China; chenruiqiu@yeah.net (R.C.); 18209858074@163.com (F.L.); huang_lng@163.com (L.H.); gdlihe0812@163.com (H.L.); 2Key Laboratory for Biodiversity Conservation in Karst Mountain Area of Southwestern China, National Forestry and Grassland Administration, Guiyang 550005, China

**Keywords:** *Dysosma*, ecological niche differentiation, distribution dynamics, habitat suitability prediction, MaxEnt modeling, climate response

## Abstract

The genus *Dysosma*, a group of perennial herbaceous plants with significant medicinal value and a relatively narrow ecological niche, is potentially at risk due to the combined pressures of habitat degradation and climate change. As their habitats continue to degrade, all species of this genus have been included in the National Key Protected Wild Plants List (Category II). Investigating the impacts of climate change on the distribution of *Dysosma* resources is vital for their sustainable utilization. In this study, the potential distribution dynamics of seven *Dysosma* species under current and three future climate scenarios (SSP126, SSP245, SSP585) were quantified using 534 occurrence points and 25 environmental variables in a MaxEnt model, accompanied by the ecological niche overlap index (Schoener’s D), dynamic metrics (relative change rate [*RCR*], change intensity [*CI*], stability index [*SI*], spatial displacement rate [*SDR*]), and centroid migration analysis. The results indicated that (1) areas of high habitat suitability were consistently concentrated in the mountainous and hilly regions of southwestern Guizhou, Chongqing, and Hubei, with the minimum temperature of the coldest month (Bio6) and the mean diurnal temperature range (Bio2) being identified as the primary driving factors. (2) The future suitable habitat areas remained highly stable overall (*SI* > 97.89%), though dynamic changes varied across scenarios. Under SSP126, only slight fluctuations were observed, with an average *CI* of approximately 3.78% and *RCR* ranging from −0.46% to 1.97%. Under the SSP245 scenario, suitable habitat areas showed a continuous, slight expansion (*RCR* = 0.45% to 1.54%), whereas under the high-emission SSP585 scenario, a typical “mid-term expansion–late-term contraction” pattern was observed, with *RCR* shifting from positive (1.32%, 1.44%) to negative (−0.92%). The *SI* reached its lowest value of 97.89% in the late term, and the spatial displacement rate increased, signaling a reorganization of the distribution pattern. (3) High ecological niche differentiation was observed within the genus, with the highest overlap index being only 0.562, and approximately one-third of species pairs exhibiting completely non-overlapping niches. (4) *Dysosma tsayuensis*, a niche-specialist species, exhibited a distribution that was highly dependent on the annual mean ultraviolet-B radiation (UVB, contribution rate 52.9%), displaying an adaptation strategy markedly different from that of conservative species. (5) Centroid analysis indicated that, although the overall centroid remained stable in Guizhou, the presence of niche-specialist species under the high-emission SSP585 scenario resulted in migration paths opposite to those observed under other scenarios. The findings reveal the potential vulnerability and differential response patterns of *Dysosma* species under rapid climate warming, thereby providing a scientific basis for targeted conservation, in situ and ex situ conservation strategies, and population restoration.

## 1. Introduction

Global climate change is profoundly reshaping the geographic distributions of species and represents one of the major threats to current biodiversity [1]. Numerous studies have demonstrated that ongoing temperature increases have driven the geographic distributions of many species to shift toward higher latitudes or elevations, with former suitable habitats progressively contracting or becoming spatially displaced [2,3,4]. The IPCC Sixth Assessment Report indicates that, under different emission scenarios, the global mean temperature by the end of this century could increase by 1.4–4.4 °C relative to pre-industrial levels [5]. The consequent increases in temperature and changes in precipitation patterns are expected to further exacerbate habitat loss and significantly affect species’ ecological traits and potential distribution ranges [6,7]. Driven by these changes, approximately one-quarter of global plant species may face extinction risk, with the habitats of many rare plants continuing to degrade [8]. Typical representatives include alpine plants highly sensitive to temperature changes [9] and regional endemic species with narrow, topographically constrained distribution ranges [10,11], whose suitable habitats are progressively contracting under climate warming. For example, studies on *Vatica guangxiensis*, an endangered tree endemic to Guangxi’s karst region, indicate that under future climate warming scenarios, its already limited suitable habitat area will be further reduced and fragmented, thereby elevating its survival risk [12]. Due to their limited ecological tolerance, rare plants are often particularly sensitive to climate warming and extreme climatic events, and their conservation remains critical for maintaining and enhancing biodiversity [8,13].Therefore, in the context of intensifying climate change and extreme events, accurately predicting changes in the suitable habitats of plant species and providing a scientific basis for in situ and ex situ conservation have become critical scientific tasks in biodiversity protection.

Species distribution models (SDMs) analyze the relationships between species distributions and environmental variables to simulate and predict spatial distribution patterns, providing an effective quantitative tool for investigating species migration and dispersal mechanisms [14]. In recent years, with the advancement of machine learning methods and ensemble modeling, approaches such as boosted regression trees (BRT), extreme gradient boosting (XGBoost), and ensemble modeling frameworks (Biomod2) have been widely applied in presence-only data modeling, demonstrating significant advantages in parameter optimization and predictive performance [15,16]. Currently, mainstream species distribution models also include mechanistic climate suitability models (Climex) [17], random forests (RF) [18], and maximum entropy (MaxEnt) models [19]. Among these methods, the MaxEnt model, owing to its low sample-size requirements, operational simplicity, and high predictive accuracy, has been widely applied and broadly acknowledged in distribution modeling studies across various biological taxa [20,21,22]. Although the MaxEnt model is widely applied, it exhibits several notable limitations: (1) it primarily relies on environmental variables such as climate and elevation and fails to adequately account for biotic interactions [23]; and (2) its predictions represent only potential suitable habitats rather than actual distribution ranges [24]. Nevertheless, it remains a commonly used tool for ecological niche prediction.

The genus *Dysosma*, belonging to the family Berberidaceae, comprises perennial herbaceous plants and currently includes nine recognized species, mainly distributed in China and Vietnam, of which seven are recorded in the Flora of China (FOC). In addition, *Dysosma villosa* has only recently been reported [25], while *Dysosma tonkinense* displays distinct taxonomic characteristics in recent morphological and phylogenetic analyses and is recognized as a separate species [26]. The majority of *Dysosma* species are mainly distributed within China’s subtropical evergreen broadleaf forest belt, with a geographic range spanning approximately 23°–32° N and 94°–122° E [27]. Previous studies suggest that China constitutes the center of species diversity and the evolutionary origin of the genus *Dysosma* [28], while the eastern Yunnan–Guizhou Plateau extending toward the Three Gorges region has historically played a pivotal role in shaping its distribution patterns and species diversification [29].The genus *Dysosma*, used in traditional Chinese medicine, not only has high medicinal value but is also among the most threatened and rare plants [30]. *Dysosma* was recorded in the *Shennong Bencao Jing Jiaozhu* [31] under the name “guijiu”, with its medicinal properties described as “effective in eliminating poisonous parasites and malevolent influences, dispelling evil energies, and detoxifying a hundred poisons.” Modern pharmacological studies have revealed that podophyllotoxin, as a key bioactive constituent, is predominantly concentrated in the rhizomes of *Dysosma* and exhibits significant antitumor, antiviral, and antibacterial activities [32]. In recent years, research on *Dysosma* has been primarily directed toward its clinical applications [32], the extraction and synthesis of its derivatives [33], and the elucidation of the mechanisms underlying its toxic effects [34,35]. However, studies on the distribution patterns and ecological suitability of *Dysosma* remain limited, and systematic investigations into the migration and dispersal mechanisms of these species are particularly lacking [36]. In the context of intensifying climate change, there is an urgent need to investigate how *Dysosma* responds and adapts to extreme climate scenarios, such as global warming, thereby elucidating its ecological adaptive potential and future survival strategies.

To clarify the influence of species-level niche differentiation on the overall distribution patterns and future prospects of the genus, this study integrated species distribution data of *Dysosma* in China and employed the MaxEnt model in combination with spatial analysis techniques to simulate the potential distribution dynamics under current and future climate scenarios, while assessing the differentiated risks among species within the genus. The objectives of this study are to: (1) reveal the patterns of niche differentiation and the key environmental drivers of *Dysosma* species across China. (2) Quantify the distribution dynamics and response trajectories of species within the genus under future climate change scenarios. (3) Assess the vulnerability and species-specific risks under climate change. (4) Identify potential conservation priority areas and management gaps within China, thereby providing a scientific basis for hierarchical protection and strategic management. The results not only reveal the patterns of niche differentiation and future risk disparities among species within *Dysosma*, but also provide a scientific basis for developing hierarchical management strategies integrating overall conservation with species-specific measures, as well as for selecting suitable sites for in situ and ex situ cultivation.

## 2. Results

### 2.1. Integration of Ecological Niches in Dysosma and Assessment of MaxEnt Predictive Performance

This study assessed the robustness of genus-level modeling across three dimensions: ecological responses, environmental space, and geographic space. In terms of ecological responses, *Dysosma delavayi*, *Dysosma difformis*, *Dysosma majoensis*, and *Dysosma versipellis* exhibit broadly similar response curves to the mean diurnal temperature range (Bio2), each following a unimodal pattern with closely aligned peaks (Appendix A). With respect to the minimum temperature of the coldest month (Bio6), *D. difformis*, *Dysosma pleiantha*, and *D. versipellis* exhibit similar response patterns (Appendix A). Only *D. versipellis* and *D. difformis* exhibited consistent response peaks across all four environmental variables, including Bio2, Bio6, annual precipitation (Bio12), and elevation (Appendix A). In the environmental space dimension, species generally clustered along PC1 and PC2; however, *Dysosm atsayuensis* and *D. versipellis* exhibited marked deviations, with their density ellipses clearly illustrating niche differentiation (Appendix A). In the geographic space dimension, the potential suitable areas of seven species overlapped with genus-level results by 31.1%, with a spatial correlation coefficient of 0.71 (Figure 1). The model achieved an AUC of 0.918 (Figure 2a), and the omission rate curve closely followed the theoretical expectation, indicating high predictive accuracy without evident overfitting (Figure 2b). Overall, genus-level modeling exhibited a certain degree of consistency with species-level distributions, although some species still showed discrepancies.

### 2.2. Overall Environmental Drivers of Dysosma Distribution

In the MaxEnt model predictions, the four environmental variables with the highest contributions were Bio6 (51.9%), Bio2 (30.5%), elevation (5.2%), and Bio12 (3.4%), collectively accounting for nearly 91% of the total contribution (Appendix A). The Jackknife test results indicated that, in single-variable analyses, Bio2 exerted the strongest influence on the distribution of *Dysosma*, followed by Bio6 (Figure 3). The results indicate that, under the current climatic conditions, the aforementioned environmental factors play a key regulatory role in the suitability of *Dysosma* growth. Based on the response curves of the key environmental variables (Figure 4), the suitable distribution areas of *Dysosma* are generally located in warm and humid climates with some tolerance to low temperatures. The optimal habitat parameter ranges are approximately 3–5 °C for Bio2 and −5–5 °C for Bio6.

### 2.3. Patterns of Niche Differentiation Among Species Within Dysosma

As shown in Figure 5, Schoener’s D index calculated across the full environmental space indicated that niche overlap among *Dysosma* species was generally low. Among 21 species pairs, seven pairs (33.3%) exhibited a niche overlap value of 0, indicating that these species did not overlap within the selected environmental space. The species pair with the highest overlap was *D. delavayi* and *D. difformis*, with a D value of only 0.562. Overall, the distribution of D values among species spanned a wide range, indicating substantial variation in niche overlap across different species pairs.

### 2.4. Key Environmental Drivers of Niche Differentiation in Dysosma

To elucidate the potential mechanisms underlying niche differentiation, comparisons were made between the key environmental drivers of the highly overlapping species pair *D. delavayi* and *D. difformis* and those of the non-overlapping pair *D. versipellis* and *D. tsayuensis*. As shown in Table 1, *D. difformis* exhibited the highest sensitivity to Bio2, with a contribution of 46.8%, whereas the contribution of this variable to *D. tsayuensis* was only 3.3%. Comparisons of environmental variable response curves between species pairs with high and low niche overlap indicated that the highly overlapping pair *D. delavayi* and *D. difformis* was primarily driven by Bio2 (Figure 6a). In contrast, *D. versipellis* and *D. tsayuensis* exhibited distinctly different driving patterns, with the distribution of *D. versipellis* mainly controlled by Bio6, whereas that of *D. tsayuensis* was predominantly influenced by annual mean ultraviolet-B radiation (UVB1) (Figure 6b).

### 2.5. Suitable Habitats of Dysosma Under Current and Future Climate Scenarios

As shown in Figure 7 and Figure 8, *Dysosma* exhibited a stable “core–satellite” distribution pattern under both current and future climate scenarios. Its highly suitable areas remained consistently concentrated in the mountainous and hilly regions of southwestern Guizhou, Chongqing, and Hubei, forming the species’ distributional core, whereas moderately and poorly suitable areas acted as satellite patches, distributed in a surrounding pattern across central Hunan, Jiangxi, Zhejiang, and central Yunnan. As shown in Table 2, under the SSP126 low-emission scenario, the total suitable area remained consistently high from the current to 2090s, ranging from 253.02 × 10^4^ km^2^ to 257.04 × 10^4^ km^2^. Under the SSP245 intermediate-emission scenario, the total suitable area showed a gradual expansion from 2050s to 2090s, increasing from 254.16 × 10^4^ km^2^ to 259.29 × 10^4^ km^2^. Under the SSP585 high-emission scenario, the area of highly suitable habitats exhibited an initial increase followed by a decrease, reaching a peak of 56.58 × 10^4^ km^2^ in 2050s, and subsequently declining to 53.83 × 10^4^ km^2^.

### 2.6. Spatiotemporal Dynamics of Dysosma Under Different Climate Scenarios

Distinct response patterns of *Dysosma* were observed under different emission pathways. Under the SSP585 high-emission scenario, suitable habitats exhibited the most pronounced changes, characterized by an initial expansion followed by a subsequent contraction. From a spatial perspective, suitable habitats expanded in southern Henan and eastern Shandong during the 2050s (Figure 9g). By the 2070s, however, these newly gained areas experienced a decline, reflecting pronounced spatial turnover (Figure 9h). By the 2090s, the region exhibited a renewed expansion trend (Figure 9i). From a mechanistic perspective, suitable habitats continued to expand during the first two periods, with relative change rates (*RCR*) of 1.32% and 1.44%, respectively. However, from the 2070s to the 2090s, a slight contraction occurred (*RCR* = 0.92%), primarily due to degradation from highly suitable to moderately suitable habitats (5.96 × 10^4^ km^2^) and from moderately suitable to poorly suitable habitats (8.39 × 10^4^ km^2^). The system’s long-term stability declined from 99.04% to 97.89%, while the spatial displacement rate (*SDR*) increased from 0.32 to 0.40, indicating that spatial instability intensified under climate change. However, under the SSP126 low-emission scenario, the system maintained dynamic equilibrium throughout the study period (current–2090s), with the stability index (*SI*) consistently at an exceptionally high level of 97.90–98.88%, and the change intensity index (*CI*) generally below 4.5%. Although local degradation from highly and moderately suitable habitats to lower suitability levels occurred during the early period (current–2050s), the overall system remained stable without significant structural changes. In the mid-to-late stages of the study, system dynamics further weakened, ultimately exhibiting a typical “quiescent-stable” pattern. The results under the SSP245 intermediate-emission scenario fell between those of the aforementioned scenarios, exhibiting a moderate expansion trend (Figure 10, Table 3).

### 2.7. Centroid Shifts of Highly Suitable Habitats of Dysosma Under Future Climate Change

As shown in Figure 11, under current climatic conditions, the centroid of highly suitable habitats of *Dysosma* is located in Sinan County, Guizhou Province (108.04° E, 28.03° N). Under the three future scenarios, although the migration directions of the centroid differ, its overall range of movement remains confined within Guizhou Province. Significance tests indicated that the centroid migration distances across all time periods were not statistically significant (*p* < 0.05) (Table 4). Under the low- and intermediate-emission scenarios (SSP126 and SSP245), the centroid primarily exhibited short-distance back-and-forth movements near northeastern Guizhou (Tongren–Dejiang region), with a maximum displacement of less than 40 km. Under the high-emission scenario (SSP585), the centroid showed a slight southward shift, but no statistically significant change was observed. Overall, regardless of the emission scenario, the centroid of highly suitable habitats of *Dysosma* remained within Guizhou Province and exhibited non-significant short-distance movements.

### 2.8. Potential Conservation Gaps for Dysosma Under Future Climate Scenarios

In this study, the current medium- and highly suitable habitats of *Dysosma* in China were designated as priority conservation areas, with a total area reaching 145.62 × 10^4^ km^2^. However, the assessment revealed a severe lack of protection coverage: only 4.44% (6.46 × 10^4^ km^2^) of the priority areas were covered by natural protected areas, leaving 95.56% as conservation gaps (Figure 12a). The suitable habitats of *Dysosma* partially overlapped with national-level protected areas in some southwestern regions (red), but the overall coverage remained low. Large contiguous areas of highly suitable habitats were not included in existing protected areas (blue), resulting in pronounced conservation gaps. Under different future scenarios, the total area of priority conservation zones generally exhibited an initial increase followed by a decline, although the overall magnitude of change remained limited. Notably, under the SSP245 and SSP585 scenarios, the area of priority conservation zones consistently exceeded that under SSP126 across all three periods (Figure 12b).

## 3. Discussion

### 3.1. Key Climatic Determinants Shaping the Geographical Distribution of Dysosma

Multiple studies have demonstrated that the MaxEnt model performs exceptionally well in ecological niche modeling, with its predictive power and accuracy often surpassing those of other approaches [37,38]. However, the full performance of the model largely depends on the quality of the input data and the appropriateness of parameter settings. Therefore, when predicting the suitable habitats of *Dysosma* using the MaxEnt model, key factors influencing model performance were systematically optimized, and the optimal parameter settings were identified using the ENMeval package (delta.AICc = 0; AUC = 0.918). Ultimately, the model yielded a highly plausible distribution pattern, with areas of high suitability concentrated from Guizhou through Chongqing to southwestern Hubei, thereby demonstrating the effectiveness of the model optimization and the robustness of its predictions. Climate change has been widely recognized as a primary driver of shifts in species geographic distributions, migration dynamics, and overall survival conditions [39]. In this context, species distribution patterns are not only the result of prolonged interactions with the environment, but also serve as detailed records of the dynamic evolutionary processes of the species [40]. Therefore, to elucidate the specific impacts of climate change on the distribution of *Dysosma*, this study simulated its potential distribution areas and identified the primary environmental drivers. The results indicate that Bio6 (contributing 51.9%) and Bio2 (contributing 30.5%) are the two primary environmental factors driving the potential distribution patterns of *Dysosma*. Diurnal temperature range (Bio2), which reflects the stability of day–night temperatures, imposes significant ecological constraints on the survival and distribution of *Dysosma* [41], a finding that is consistent with previous studies. The suitable ranges of minimum temperature of the coldest month (Bio6: −5 to 5 °C) and diurnal temperature range (Bio2: 3 to 5 °C) jointly define the unique ecological niche of *Dysosma*, which occurs in warm-temperate to mid-subtropical mountainous regions that avoid extreme cold (Bio6 > −5 °C) while still experiencing moderate winter temperatures (Bio6 < 5 °C). The climate niche revealed by the model is highly consistent with microhabitat preferences reported in the literature, collectively indicating that this group is a shade-tolerant herb strictly adapted to subtropical montane forest environments [27]. Furthermore, the strict requirement of *Dysosma* for a narrow diurnal temperature range (Bio2: 3–5 °C) highlights its reliance on the stable microclimate maintained by forest canopies, a finding that is consistent with large-scale microclimate observations [42]. Physiological studies provide further evidence for this: Palaniyandi & Jun [43] reported that *Dysosma* exhibits more vigorous growth under low-temperature conditions (4–6 °C), with biomass, chlorophyll, carotenoid, and toxin contents significantly higher than those of plants grown under greenhouse conditions at 25–30 °C. These findings further confirm, at the physiological level, the critical role of temperature conditions in regulating the growth and development of *Dysosma*. However, the distribution of *Podophyllum hexandrum*, another member of the Berberidaceae family, is more dependent on precipitation of the driest month (Bio14) and mean annual temperature (Bio1) [44]. This stark contrast in dominant environmental factors highlights the decisive role of life form and habitat preference in shaping species distributions [45]: *Dysosma*, as a shade-tolerant understory herb, relies on a stable microclimate that avoids extreme low temperatures, whereas *P. hexandrum*, a relatively sun-loving species, responds more directly to large-scale climatic patterns of moisture and temperature, thereby establishing markedly distinct ecological niches for the two species. Therefore, species-specific ecological strategies must be fully considered when predicting the impacts of climate change. For *Dysosma*, winter warming may create opportunities for expansion into higher latitudes or elevations; however, climate change not only involves increases in mean annual temperature but also entails alterations in extreme winter cold and the degradation of critical microclimates due to forest fragmentation, all of which could pose serious threats to the long-term survival of the species [46].

Although elevation, as a composite geographic factor integrating temperature, precipitation, and solar radiation, plays a fundamental role in shaping species distribution patterns [47] and is often regarded as a primary driver of predictive differences [48], its contribution to the distribution of *Dysosma* in the present study was relatively low. This pattern aligns with general principles in species distribution modeling: when climate drivers that are more directly linked to species physiology (e.g., Bio6, Bio2) are incorporated into the model, their explanatory power is preferentially assigned to these direct factors rather than to topographic variables serving as proxies [49,50]. Therefore, the independent contribution of elevation is diminished, indicating that variables such as Bio6 and Bio2 more accurately capture the key physiological thresholds constraining the distribution of *Dysosma*. Furthermore, moderate slopes ensure adequate drainage, effectively preventing root rot caused by water accumulation under the forest canopy [51], which aligns closely with the hydrophilic yet water-sensitive ecological preference of *Dysosma*. Therefore, at the fine-scale habitat level, the regulatory effect of topography on moisture conditions may hold more direct ecological significance than regional annual precipitation, highlighting the unique water-use strategy of *Dysosma* [52]. However, while genus-level predictions provide an overall pattern, they may, to some extent, obscure finer-scale ecological adaptations within *Dysosma*. Even when major climatic factors dominate overall distribution patterns, individual species may possess distinct ecological thresholds and response mechanisms to temperature, moisture, or microtopographic conditions, resulting in high niche differentiation [53]. At the broader subfamily level of Podophylloideae, studies by Ye Wenqing, Zhu Shanshan, and colleagues [54] have demonstrated significant niche differentiation among genera and geographic populations, primarily driven by key climatic factors such as temperature seasonality. This finding provides a broader evolutionary perspective on the continuum of niche differentiation observed among *Dysosma* species, ranging from complete separation to moderate overlap, and suggests that this group may generally exhibit diverse ecological adaptation patterns.

### 3.2. Patterns of Niche Differentiation and Evolutionary Adaptation Within Dysosma Under Current Climatic Conditions

In this study, the niche overlap index Schoener’s D among seven *Dysosma* species was calculated using the maximum value method based on gradients of 25 environmental variables. The results indicate a high gradient of niche differentiation within *Dysosma*: ranging from near-zero overlap between species such as *D. versipellis* and *D. delavayi* or *Dysosma aurantiocaulis*, to moderate overlap between species such as *D. difformis* and *D. majoensis* or *D. pleiantha*, thereby forming a continuum of niches from complete separation to partial overlap. This pattern, quantified based on niche overlap, directly illustrates the high heterogeneity in niche relationships among closely related species. Some species pairs, such as *D. delavayi* and *D. difformis*, exhibit niche conservatism, sharing the mean temperature of the driest season (Bio2) as a key driving factor, with highly congruent response curves (Figure 6a). In contrast, other species pairs show either high niche separation or moderate overlap, likely resulting from a combination of recent divergence, gene flow, and other factors [55], the precise mechanisms of which remain to be elucidated. Similar frameworks based on niche overlap have been demonstrated to provide an effective tool for quantifying interspecific relationships in studies of other endangered plant species. For example, studies on the endangered tree species *Zelkova serrata* and its associated co-occurring species have shown that climate change may induce dynamic alterations in patterns of niche overlap [56]. This finding echoes the continuous spectrum of niche differentiation observed within *Dysosma* in the present study, providing a comparative perspective for further investigation of its underlying mechanisms and interspecific interactions.

In contrast to the niche-conservative pattern observed in *D. delavayi* and *D. difformis*, *D. tsayuensis* may exhibit niche innovation. Within *Dysosma*, taking *D. versipellis* and *D. tsayuensis* as examples, the distribution of the former is primarily driven by Bio6, whereas that of the latter mainly depends on UVB1 (Figure 6b). This fundamental shift in key driving factors reflects a strong signal of adaptive evolution [57]. The adaptation of *D. tsayuensis* to high-altitude, high-radiation environments may represent a pivotal evolutionary innovation enabling the genus to colonize novel habitats. This adaptation may manifest phenotypically as alpine-plant-like photosynthetic regulation, secondary metabolite accumulation, and antifreeze protein expression [58], while at the molecular and genetic levels it reflects systemic reorganization, with genomic analyses indicating convergent evolutionary patterns across species. For instance, Zhang et al. [59] found through analyses of gene selection pressure and protein evolutionary rates that genes or pathways associated with self-incompatibility, cell wall modification, DNA repair, and stress resistance exhibited convergent evolution, providing a critical genetic foundation for alpine plants to adapt to extreme cold, high ultraviolet radiation, and hypoxic environments. The closely related genus *Diphylleia*, particularly *Diphylleia sinensis*, occurs in shaded understories or along streams at mid to high altitudes (1880–3700 m) in China. Although it continues to prefer humid and shaded habitats, the ultraviolet radiation in these mountainous regions is substantially higher than in lowland areas [60], and UV-B as well as related light stress factors are likely more influential in determining its distribution than for low-altitude, closely related *Dysosma* species. This intensified UV-B stress may have driven the evolution of corresponding photoprotective mechanisms, such as the accumulation of secondary metabolites [61]. Furthermore, multiple studies within the family Berberidaceae have demonstrated that light environments, including light quality and UV-B radiation, play a critical role in adaptive differentiation among genera. For example, short-term UV-B exposure has been shown to significantly alter leaf morphology and increase the content of bioactive compounds in *Epimedium brevicornu* [62]. Transcriptome-based studies have further revealed candidate genes associated with local adaptation among different populations of the genus *Epimedium* [63]. Additionally, treatments with varying light qualities (e.g., red, yellow, and blue light) have been demonstrated to significantly influence flavonoid secondary metabolite accumulation in these species [64]. Therefore, responses of Berberidaceae plants to light environmental pressures, particularly combined UV-B and light quality signals, are phylogenetically conserved yet capable of independent and repeated evolution. This is highly consistent with the substantial shift observed in *D. tsayuensis* within this study, where its ecological niche drivers transition from temperature to dimensions of light intensity and light quality, further supporting an adaptive evolutionary trajectory driven by strong environmental selective pressures. These multilayered adaptive mechanisms provide critical insights into the ecological niche differentiation and potential evolutionary strategies of *D. tsayuensis* in high-altitude environments.

In summary, multiple Berberidaceae lineages distributed from mid- to high-altitude environments, including *Dysosma*, *Diphylleia*, and *Epimedium*, exhibit similar adaptive trends. As species colonize harsher montane or alpine habitats from relatively mild lowland environments, the core drivers of their survival strategies often undergo fundamental restructuring. These drivers shift from traditional climatic factors such as temperature and moisture to a novel adaptive framework dominated by ultraviolet radiation (UV-B), light quality composition, and a suite of extreme physical stressors. This cross-lineage “shift in core adaptive drivers” essentially represents a repeatable evolutionary response exhibited by organisms under intense and highly directional environmental selection pressures [65]. It demonstrates that adaptive evolution involves not only incremental adjustments of existing traits but may also entail qualitative transformations or restructuring of core survival strategies. Ultimately, within the same genus-level framework, different species achieve high niche differentiation and geographic segregation through differential specialization across both ancestral and novel environmental dimensions, thereby maintaining the diversity and coexistence patterns of the clade within montane and hilly landscapes.

### 3.3. Predicting Future Climate Suitability of Dysosma Species Based on Ecological Niche Modeling

The multidimensional indicator framework developed in this study demonstrated high applicability in analyzing the spatiotemporal dynamics of *Dysosma’s* suitable habitats, successfully revealing its patterns of change under different climatic scenarios. *Dysosma* is projected to remain highly stable under future climate scenarios (*SI* 97.89–99.98%), with its dynamic patterns exhibiting a gradient response to emission intensity: remaining stable under SSP126, experiencing gradual expansion under SSP245, and showing an initial expansion followed by contraction under SSP585, consistent with the dose–response relationship of climatic forcing [66]. During the early phase of change under the SSP126 scenario (current to 2050s), high *CI* values combined with moderate *SDR* values indicate that the system is in an active “turnover” state, consistent with the ecological niche migration theory [67]. However, during the late phase of the study under the SSP585 scenario (2070s–2090s), the system dynamics underwent a critical shift: although *CI* values remained high, *SDR* values showed a decreasing trend, indicating a depletion of expansion potential. Consequently, the system exhibited a “net contraction” pattern, characterized by the degradation of high-suitability areas to lower levels (Figure 8c), likely reflecting that the intensifying climate change has exceeded the ecological niche tolerance limits of the species [68]. This “stability-within-dynamics” pattern differs from the predictions reported by Fan et al. [69] for Magnoliaceae tree species, which exhibited strong shifts in their distribution ranges. This discrepancy may be attributed to the strong microclimatic dependence of *Dysosma* as an understory herb, which limits its ability to rapidly shift in response to changes in macroclimatic averages.

The spatial distribution results indicate that the high-suitability areas of *Dysosma* are primarily concentrated in the mountainous and hilly regions of southwestern Guizhou, Chongqing, and Hubei (Figure 9), closely overlapping with biodiversity hotspots and historical refugia such as the Wuling and Dalou Mountains [70]. This stable distribution may be related to the interactions among topography, ecosystems, and evolutionary history. Complex terrain not only buffers extreme low temperatures [71] but also creates diverse microclimatic units, such as valleys and slope aspects, providing microenvironmental heterogeneity for *Dysosma*. Well-preserved forest systems rely on these topographic features to maintain local water availability and humid microclimates, thereby fulfilling the species’ hygrophilous requirements [71]. Furthermore, the complexity of topography may also play a role at the evolutionary level: landscape genomics studies have shown that such mountainous systems provided refugia for plants (e.g., Tetrastigma hemsleyanum) during historical climatic fluctuations, while promoting local adaptation and maintaining genetic diversity, thereby enhancing the adaptive potential of populations to future environmental changes [72]. Centroid analysis indicated that the distribution centroid of *Dysosma* has remained largely stable within Guizhou, suggesting that this region may serve as a key climatic refugium (Figure 11). The results indicate that the stability observed in Guizhou is not attributable to a single factor, but rather arises from the combined effects of topography, climate, and ecological history. Furthermore, the newly suitable areas are primarily concentrated in southern Henan and northeastern Shandong, with scattered distributions in central Yunnan, northern Anhui, and northern Jiangsu, showing an overall trend of eastward migration toward central and eastern China. This finding is consistent with the results of Huang et al. [41], collectively indicating the potential directions of *Dysosma* expansion under climate change. This consistency not only enhances the reliability of the model results but also highlights the potential of areas surrounding the Huang-Huai-Hai Plain as future suitable habitats for the genus *Dysosma*. However, many of these newly identified suitable areas are situated within agricultural plains experiencing intensive human activity, and whether the existing forest cover and moist habitat conditions can support successful colonization by the species remains to be confirmed through field verification. Moreover, the formulation of more species-specific conservation or in situ and ex situ strategies necessitates moving beyond the aforementioned general predictions derived solely from climatic factors.

Notably, the pronounced niche differentiation within the genus *Dysosma* (Figure 5) suggests that genus-level suitability predictions could generate systematic biases across species. Centroid shift trends predominantly reflect the overall patterns of species with similar niche requirements and relatively “conservative” ecological strategies (e.g., *D. delavayi* and *D. difformis*), but may fail to adequately capture specialized species that undergo niche “dimensional shifts” (e.g., *D. tsayuensis*). The distribution of the latter strongly depends on UVB1 and the high-altitude, high-radiation habitats it represents, rendering it highly sensitive to climate change and susceptible to the typical “ceiling effect” under ongoing warming [73]. As a result, the rate of habitat loss and the corresponding extinction risk may substantially exceed the average values predicted by genus-level models. Nevertheless, the dynamic change indicators developed in this study continue to offer tangible spatial guidance for conservation planning. Stable high-suitability areas (Figure 9) can be considered core “refugia” for sustaining the diversity of the genus *Dysosma* and should be prioritized for protection within conservation networks. Centroid shift trajectories and potential expansion areas identify key corridors where ecological connectivity should be maintained or restored, thereby facilitating *Dysosma* species—especially those with relatively conservative niches—to implement necessary distributional adjustments under future environmental change. Simultaneously, while the overall centroid of the genus *Dysosma* remains in Guizhou across all three SSP scenarios, the migration trajectories diverge markedly—moving toward higher latitudes under SSP126 and SSP245, but toward lower latitudes under the high-emission SSP585 scenario. This anomalous pattern may indicate that specialized species favoring high-altitude, cool environments (e.g., *D. tsayuensis*) cannot “escape” to higher elevations under extreme warming, thereby exhibiting model-predicted contractions toward lower latitudes. Such high sensitivity to climate warming is relatively widespread among high-altitude-dependent species within the Berberidaceae, although the precise trajectories of their distributional responses may differ substantially due to variations in niche composition. For example, model projections for *P. hexandrum*, a member of the same family, indicate that its suitable habitats are projected to contract toward higher elevations and decline markedly under climate warming [44]. In contrast, our results suggest that *D. tsayuensis*, which similarly favors high-altitude environments within the genus *Dysosma*, may exhibit a contraction of suitable habitats toward lower elevations. Although these patterns proceed in opposite directions, they both illustrate the common challenges encountered by high-altitude specialist species in the context of climate warming, with their core habitats potentially experiencing significant shifts and facing a pronounced risk of loss. Thus, for niche-specialized species within *Dysosma*, the rate of future habitat loss and the corresponding extinction risk is likely to be underestimated by models relying on genus-wide average trends or assumptions of uniform elevational responses.

### 3.4. Targeted Conservation Strategies for Dysosma

The identification and strategic delineation of priority conservation areas for *Dysosma* under current and future climate scenarios are essential to secure its long-term persistence. However, the gap analysis indicates that existing protected areas encompass only about 4.44% of the identified priority zones (approximately 6.46 × 10^4^ km^2^), leaving as much as 95.56% unprotected. Moreover, these gaps are not evenly distributed but are instead concentrated in key southern regions, including Guizhou, Sichuan, and Hunan (Figure 12a). In particular, core gap regions such as the Wuling and Dalou Mountains at the junction of Guizhou, Chongqing, and Hubei exhibit minimal overlap with the current protected-area network, indicating that existing conservation frameworks are insufficient to encompass the most critical and climatically stable habitats of *Dysosma*. The findings of this study align with those of Zhang et al. [74] on the distribution of priority conservation areas for nationally protected wild plants, thereby underscoring the strategic importance of the mountainous and hilly belt extending across southwestern Guizhou, Chongqing, and Hubei for future conservation planning. Meanwhile, this region represents a biodiversity hotspot, suggesting that the designated priority conservation areas may play a crucial role under future climate change [75]. Therefore, conducting climate-change monitoring within these priority conservation areas and assessing the sensitivity of *Dysosma* to future climatic shifts become particularly important. However, conservation efforts cannot be postponed until sufficient information on wild plant protection becomes available, as delays in action may result in missed opportunities to safeguard vulnerable species [76]. To mitigate the risk of losing conservation efficacy due to climate change [77], it is essential to balance both the effectiveness and timeliness of conservation actions.

Drawing on the aforementioned mechanisms of niche differentiation and projected future risk patterns, corresponding conservation strategies can be proposed; nevertheless, these recommendations are based on model predictions and inherently involve a degree of uncertainty. This study adopts a “full-dispersal” assumption, which does not adequately account for the dispersal constraints of *Dysosma* in natural ecological processes. The capacity of plants to track climate change is constrained not only by potential dispersal distances, but also by additional factors including microhabitat filtering, seedling establishment rates, dispersal vector efficiency, and habitat fragmentation [78]. For instance, species like *Rhexia virginica*, which have successfully expanded owing to strong dispersal capabilities [79], highlight the widespread risks confronting dispersal-limited species. *Dysosma* species are slow-lived, perennial herbs with limited dispersal distances; their dispersal fronts are strongly constrained by forest structure, humidity, and interspecific competition, and their migration rates may lag considerably behind the rapid shifts of climatically suitable habitats [80]. Consequently, a considerable fraction of model-predicted future climatically suitable areas may be “suitable yet inaccessible” in reality, resulting in an underestimation of species’ true contraction risk. Moreover, the representativeness of key environmental variables may also affect the accuracy of predictions. Future land-use changes and habitat fragmentation were not incorporated into the models [81], implying that actual available habitats may be smaller than predicted. Soil properties, including pH, texture, and organic matter, are critical for microhabitat selection by *Dysosma*, yet global-scale soil datasets (e.g., HWSD) poorly capture local heterogeneity and exhibit collinearity with climate and topographic variables at regional scales (Appendix A), thereby limiting the reflection of their independent contributions (Appendix A). Additionally, the models did not incorporate biotic interactions, including pollination and competition, potentially resulting in an overestimation of population establishment and persistence under novel climatic conditions [82]. Even under climatically suitable conditions, the lack of ecological interactions may hinder populations from successfully establishing or persisting within potential suitable habitats.

To enhance predictive accuracy, future studies could integrate dispersal models with habitat connectivity analyses to simulate more realistic migration scenarios, incorporate high-resolution land-use change data, and, at fine spatial scales (e.g., watersheds or slopes), combine in situ soil surveys with physiological and ecological experiments to quantify the influence of soil and microenvironmental factors on habitat suitability, thereby developing more mechanistic distribution models. Despite these limitations, quantitative analyses of potential suitable habitats for *Dysosma* still provide a critical foundation for formulating scientifically informed, tiered conservation strategies. Accordingly, the present study outlines the following priority conservation strategies: (1) Prioritize the protection of stable suitable areas. Model-identified current and future long-term high-suitability areas (Figure 9) should be incorporated into priority conservation zones, management of existing protected areas should be strengthened, and fragmentation of core habitats should be minimized. (2) Establish ecological corridors. In key expansion zones along centroid migration pathways, corridors should be implemented through vegetation restoration and the creation of small protected areas, thereby facilitating potential migration for niche-conservative species, including *D. difformis* and *D. delavayi*. (3) Priority monitoring of specialized species. For niche-divergent, specialized species, such as *D. tsayuensis*, long-term monitoring stations should be established in high-altitude refugia to track key environmental indicators and, when necessary, implement assisted migration. (4) Strengthen ex situ conservation. Living gene banks of *Dysosma* should be established, and ex situ cultivation should be implemented in experimental sites simulating future climates or varying elevations to assess environmental tolerance and mitigate pressure on wild populations. (5) Strengthen long-term monitoring and mechanistic research. A monitoring network spanning major distribution areas should be established, integrated with functional trait and genomic studies to elucidate the physiological and genetic underpinnings of niche differentiation. (6) Strengthen institutional support. Relevant laws and regulations should be reinforced to provide policy backing for the long-term conservation and sustainable use of *Dysosma*. The findings of this study provide a scientific basis for the conservation, domestication, and sustainable use of *Dysosma* species.

## 4. Materials and Methods

### 4.1. Data Sources and Processing of Dysosma Spatial Distribution

This study used the genus *Dysosma*, as defined by the *Flora of China* (FOC) taxonomic system, as the research unit, focusing on seven species: *Dysosma aurantiocaulis*, *Dysosma difformis*, *Dysosma majoensis*, *Dysosma pleiantha*, *Dysosma tsayuensis*, *Dysosma delavayi*, and *Dysosma versipellis*. In addition, because *Dysosma villosa* is known from only a single type specimen record, it did not meet the requirements for constructing a stable and reliable MaxEnt model and was therefore excluded from the modeling analyses.  Species occurrence data were obtained from the Chinese Virtual Herbarium (CVH, http://www.cvh.ac.cn/), the Global Biodiversity Information Facility (GBIF, http://www.gbif.org/), and field surveys conducted in situ. For records with locality information specified at the village or town level, geographic coordinates were extracted using Google Earth Pro version 7.3.6 (Google LLC, Mountain View, CA, USA). For records documented only at the county level or with incomplete information, georeferencing was conducted within a plausible elevation range based on specimen altitude data. A total of 782 occurrence records with precise geographic coordinates were obtained through the aforementioned procedures (Table 5). To prevent model overfitting caused by duplicate records within the same grid cells, spatial thinning was performed at both the genus and species levels using the R package ENMTools 4.5.1. Spatial thinning was based on 5 km × 5 km environmental variable grids. Since the grid boundaries of all environmental variables were aligned at this resolution, using any single variable for thinning produced identical results. After processing, 534 occurrence records were retained at the genus level (Figure 13), and 598 records were retained at the species level (Table 6).

### 4.2. Selection of Environmental Variables for Dysosma

To identify the main environmental drivers of *Dysosma*, a total of 60 initial environmental variables (Appendix A) were selected as preliminary inputs for model screening and construction. The climatic variables included both current climate conditions (1970s–2000s) and future projected scenarios, comprising 19 bioclimatic variables and digital elevation model (DEM) data, all obtained from the WorldClim dataset [83] (http://www.worldclim.org). Slope and aspect were derived from DEM data using ArcGIS 10.2 (Esri, Redlands, CA, USA). Thirty-two soil variables were obtained from the Chinese soil dataset v1.1 of the Harmonized World Soil Database (HWSD, http://vdb3.soil.csdb.cn), and six ultraviolet (UV) variables were sourced from the Global UV-B Radiation Database [84] (http://www.ufz.de/gluv/, accessed on 16 December 2025).

The modeling procedure was conducted in four sequential steps. (1) An initial MaxEnt v3.4.4 (Princeton University, Princeton, NJ, USA) run was performed using distribution points following redundancy removal to estimate the contribution rates of 60 environmental variables. During this step, the MaxEnt options “Create response curves”, “Make pictures of predictions”, and “Do jackknife to measure variable importance” were enabled. The output format was set to Logistic, the file type to .asc, and the number of replicates to 10, while all other settings were retained as default. (2) Environmental variables with zero contribution were first removed, and the remaining variables were extracted at species occurrence points using ArcGIS “Sampling” tool, after which highly correlated variables (|r| ≥ 0.80) were excluded based on Pearson correlation analysis (Appendix A) [85]. Although Bio14 was highly correlated with Bio12 at the genus level (r = 0.83, Appendix A) and at the species level (*D. pleiantha*, r = 0.86, Appendix A), it was retained due to its ecological significance and relatively high contribution in preliminary model construction: 17.1% at the genus level (Appendix A) and 48% at the species level (*D. pleiantha*, Appendix A). (3) The ENMeval tool (version 2.0.0) was employed to optimize the regularization multiplier (RM) and feature combination (FC) parameters for the environmental variables retained after removing highly correlated predictors, thereby identifying the optimal modeling configuration. (4) The MaxEnt model was rerun using the optimized parameters and the environmental variables retained after removing highly correlated predictors, generating final suitability prediction rasters in Logistic format, environmental response curves, and the final contribution rates of all variables (Appendix A). Sensitivity analyses of the robustness of predicted suitable areas for *Dysosma*, along with calculations of environmental niche overlap indices, were conducted using the 25 environmental variables optimized at the genus level.

### 4.3. Future Climate Data and Scenario Settings

To simulate potential distribution patterns of *Dysosma* under future climate change scenarios, downscaled CMIP6 climate data from WorldClim v2.1 at a spatial resolution of 2.5′ were utilized, ensuring consistency with the baseline climate data (1970–2000). Future climate projections were generated using the BCC-CSM2-MR general circulation model (GCM) from CMIP6, developed by the National Climate Center of the China Meteorological Administration and widely applied in climate change impact assessments at global and regional scales [86,87,88,89,90]. The Shared Socioeconomic Pathways (SSPs), the core scenario framework of CMIP6, define future socioeconomic development trajectories by quantifying key variables including population, GDP, urbanization, inequality, and energy structure, and are employed to assess comprehensively the potential impacts of socioeconomic changes on the climate system [91,92]. To capture climate uncertainty across different emission pathways, three representative scenarios were selected: low-emission SSP1-2.6, medium-emission SSP2-4.5, and high-emission SSP5-8.5. Future climate data covered three time periods: the 2050s (2041–2060), the 2070s (2061–2080), and the 2090s (2081–2100). For all future scenarios, the 19 bioclimatic variables were derived from the downscaled BCC-CSM2-MR products provided by WorldClim and employed in MaxEnt models to predict the potential suitable areas of *Dysosma* under future climate conditions and to analyze their spatiotemporal dynamics.

### 4.4. Performance Evaluation and Parameter Optimization of the MaxEnt Model

Model fit is influenced by the regularization multiplier (RM) and the feature combination (FC) parameters. This study used the ENMeval package in R 4.5.1 (R Core Team, Vienna, Austria) to optimize model parameters for selecting the best-fitting model: the regularization multiplier (RM) was set from 0.5 to 4, with increments of 0.5, and six feature combinations (FCs) were tested: L, H, LQ, LQH, LQPH, and LQPHT. The Akaike Information Criterion correction (AIC) was used as a measure of model fit [93], with a value of 0 indicating that the model parameters enhance both accuracy and scientific validity. Additionally, the performance of the MaxEnt model was assessed using the area under the receiver operating characteristic curve (AUC-ROC) [94]. An AUC value ranging from 0.5 to 0.7 indicates poor performance, from 0.7 to 0.9 indicates moderate performance, and greater than 0.9 indicates high performance [95]. As shown in Table 7, to enhance model robustness across varying data scales, the genus-level model employed Subsample (75% training, 25% testing), while the species-level model utilized stratified sampling based on sample size (*n* ≥ 50: 70%/30%; 20 ≤ *n* < 50: 60% training; *n* < 15: Bootstrap).

### 4.5. Model Robustness Validation and Niche Differentiation Assessment Within Dysosma

To evaluate the validity of genus-level modeling and assess potential biases arising from interspecific ecological differences, this study performed a consistency analysis at both genus and species scales across three dimensions: ecological response, environmental space, and geographic space. First, an overall assessment was performed to evaluate the ecological niche conservatism of species within the genus, thereby providing a scientific basis for genus-level modeling. Using the four most influential environmental variables from the genus-level model, univariate response curves for each species were extracted (*_only.csv). Given the variability in species’ responses to dominant environmental factors, the number of species represented in each response curve differs. By overlaying the response curves of species that exhibit significant responses, the similarity in their ecological responses across critical environmental gradients is assessed. Next, by identifying species pairs with extreme levels of overlap (either high or low) in the niche overlap matrix, potential niche specialization and highly similar species assemblages are examined, thereby unveiling the specific patterns of niche differentiation within the genus. Given the current climatic conditions, environmental variables at the genus level corresponding to species distribution points were extracted from environmental grids. The Schoener’s D index was calculated using the ecospat package in R to quantify the pairwise ecological niche overlap between species [96].The calculation formula is as follows:$$D=1-\frac{1}{2}\sum_{i=1}^{n}\left|p_{x,i}-p_{y,i}\right|$$ where p_(x,i) and p_(y,i) represent the normalized environmental variable densities of species x and y at environmental grid point i, ∑*(i = 1)^n^ p*(x,i) = ∑*(i = 1)^n^ p*(y,i) = 1; where n is the total number of grid points. To ensure comparability across species with varying sample sizes, kernel density estimation was consistently applied using a 35 × 35 grid [97,98], which provides stable density estimates for the species with the smallest sample size, *D. aurantiocaulis* (*n* = 8), while preventing the loss of critical distribution information due to excessively coarse grids. The ecological niche distributions of the species were subsequently visualized in the PCA environmental principal component space. Finally, species-level models were developed based on 25 environmental variables selected by MaxEnt, and the maximum value method [99,100] was used to integrate predictions for the seven species into a species-wide suitable habitat map for comparison with the genus-level prediction map. The Pearson correlation coefficient and the overlap ratio of high suitability areas (*p* ≥ 0.5) were calculated using the raster package in R 4.5.1 to quantify the consistency of predictions across both spatial scales.

### 4.6. Classification of the Potential Distribution Patterns of the Genus Dysosma

The Habitat Suitability Index (HSI) can effectively quantify the potential support provided by environmental factors for species distribution, making it an important tool for scientifically assessing species habitat suitability and quality [101]. Reclassification of the ASCII files from the MaxEnt output was performed using ArcGIS, using four potential habitat types: highly suitable habitat (0.5 ≤ *p* ≤ 1.0), moderately suitable habitat (0.3 ≤ *p* < 0.5), low suitability habitat (0.1 ≤ *p* < 0.3), and unsuitable habitat (*p* < 0.1).

### 4.7. Construction of Dynamic Indicators for Changes in the Suitable Habitat of the Genus Dysosma

In this study, the initial area was defined as the total area of all habitat suitability levels (low, moderate, and highly suitable habitats). As shown in Table 8, to construct a multidimensional evaluation framework, four indicators were used to represent the four core attributes of change. (1) Relative Change Rate (*RCR*) (magnitude of change); (2) Change Intensity (*CI*) (activity level of change); (3) Stability Index (*SI*) (core area persistence); (4) Spatial Displacement Rate (*SDR*) (spatial reorganization pattern). Although some indicators (e.g., Relative Change Rate, Stability, and Spatial Displacement Rate) are interrelated in their calculations (Figure 14), they offer different perspectives on the change process, collectively forming a comprehensive analytical framework. Subsequent analyses will integrate all indicators to avoid biased conclusions arising from the limitations of any single one.

### 4.8. Analysis of Centroid Shift in the Distribution of the Genus Dysosma

To reveal the direction and spatial extent of changes in the suitable habitat of the genus *Dysosma*, this study calculated the migration paths and distances of the geometric center (centroid) of the suitable habitat across different periods. First, the highly suitable habitat areas were extracted in ArcGIS, and the centroid positions of the genus *Dysosma* in China were calculated based on their spatial distribution for each period. This allowed the simulation of centroid migration trajectories and quantification of migration distances under different climate scenarios. Next, the Kruskal–Wallis test was performed in SPSS 22 (IBM Corp., Armonk, NY, USA), with a sample size of *n* = 3 per group, to compare differences between different scenarios. Given the extremely small sample size and the difficulty in meeting the normality assumption, non-parametric statistical methods were used to compare the groups to enhance the robustness of the results.

### 4.9. Analysis of Conservation Gaps for the Genus Dysosma

In this study, the current moderate and highly suitable habitats of the genus *Dysosma* in China were designated as priority conservation areas, with the boundaries of China’s nature reserves serving as the limits. The “Extract by Mask” tool in ArcGIS was used to extract the potential suitable habitats of the genus *Dysosma* for different periods, and the dynamic changes in the protected area over time were quantified.

## 5. Conclusions

In this study, the optimized MaxEnt model was used to simulate the distribution of suitable habitats for the genus *Dysosma* under various climate scenarios. The results of this study indicate that the lowest temperature of the coldest month (Bio6) and the mean daily temperature range (Bio2) are the key environmental factors driving the distribution pattern of the genus *Dysosma*. Under the current climatic conditions, its potential suitable habitats are primarily found in the subtropical montane forests of southern China, where the complex terrain in regions such as Sichuan, Chongqing, and Guizhou creates forest microclimates, providing essential climate refuges for the species. Under future climate scenarios, the area of suitable habitats remains relatively stable under the SSP126 pathway. However, under the SSP245 and SSP585 pathways, as temperatures rise, suitable habitats first expand slowly and then gradually contract, suggesting that climate warming may reshape the species’ potential distribution patterns. Niche pattern analysis reveals significant differentiation among species within the genus *Dysosma*, with the highest ecological niche overlap index (D) value being only 0.562. The key driving factor for the specialized species *D. tsayuensis* is UV-B radiation during the growing season, contributing 52.9%, which reflects its unique adaptive strategy evolved in contrast to niche-conservative species. Centroid migration analysis indicates that the centroid of the suitable habitat remains generally stable within Guizhou. However, under the SSP585 high-emission scenario, significant reverse migration occurs, further suggesting that genus-level predictions may obscure the individual responses of species with distinct niche differentiation. From the perspective of conservation status, significant gaps exist in the protection of regions such as Guizhou and Chongqing, while the protected areas in southwest China are limited and poorly connected. This situation poses a threat to the long-term survival of the genus *Dysosma*. For niche-specialized species like *D. tsayuensis*, targeted monitoring and protection should be prioritized. The results of this study provide a scientific basis for the formulation of conservation strategies for the genus *Dysosma* under global climate warming, as well as for resource management and sustainable utilization.

## Figures and Tables

**Figure 1 plants-15-00162-f001:**
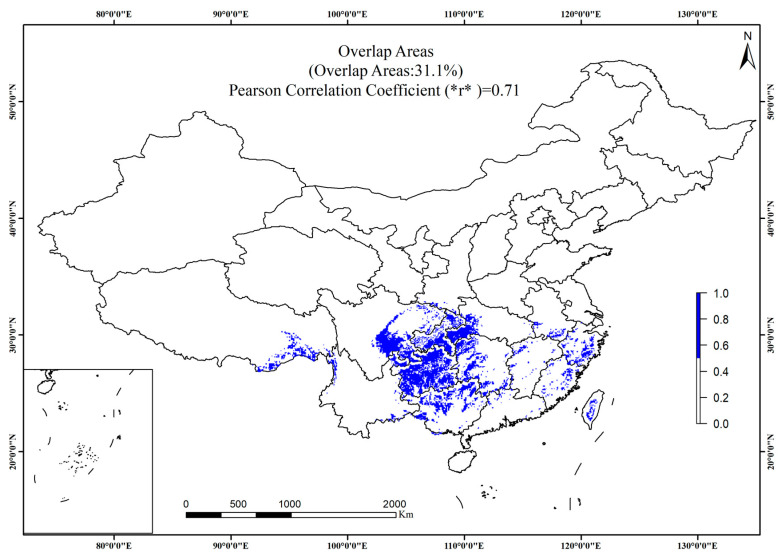
Geographic overlap of potential suitable areas for *Dysosma*.

**Figure 2 plants-15-00162-f002:**
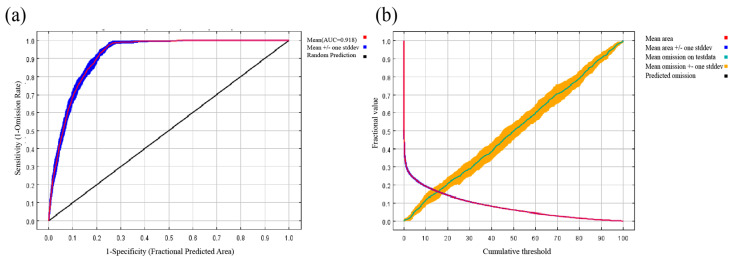
Validation of MaxEnt model predictions for *Dysosma*: (**a**) omission rates; (**b**) receiver operating characteristic (ROC) curve.

**Figure 3 plants-15-00162-f003:**
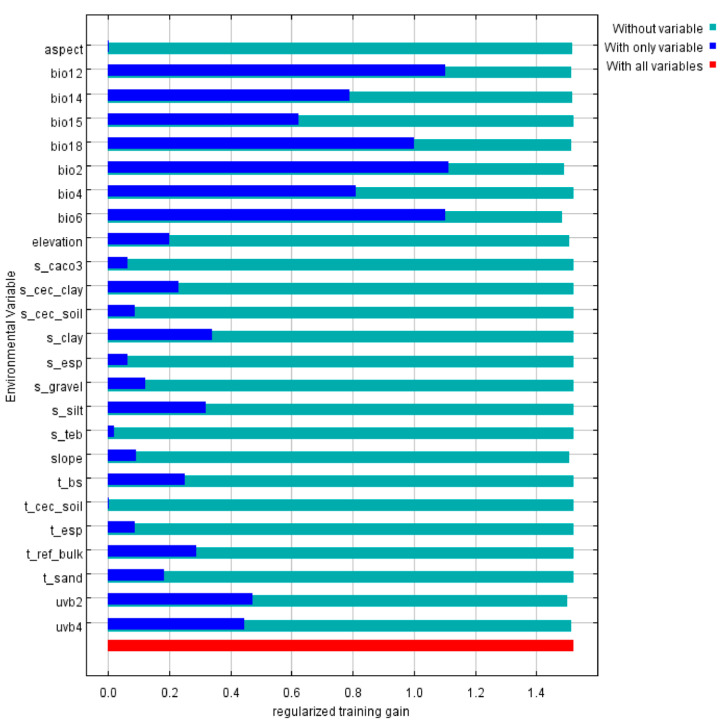
Contribution Rates of Regularized Training Gain.

**Figure 4 plants-15-00162-f004:**
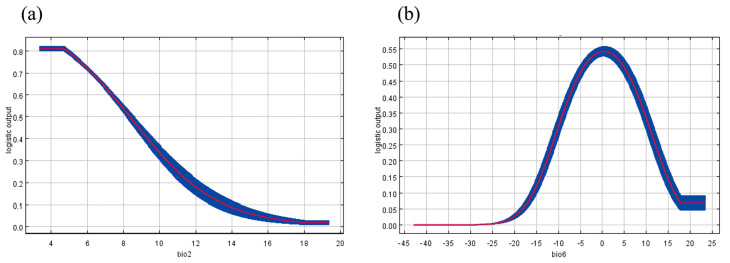
Ecological response curves of *Dysosma* to key environmental variables: (**a**) Bio2; (**b**) Bio6.

**Figure 5 plants-15-00162-f005:**
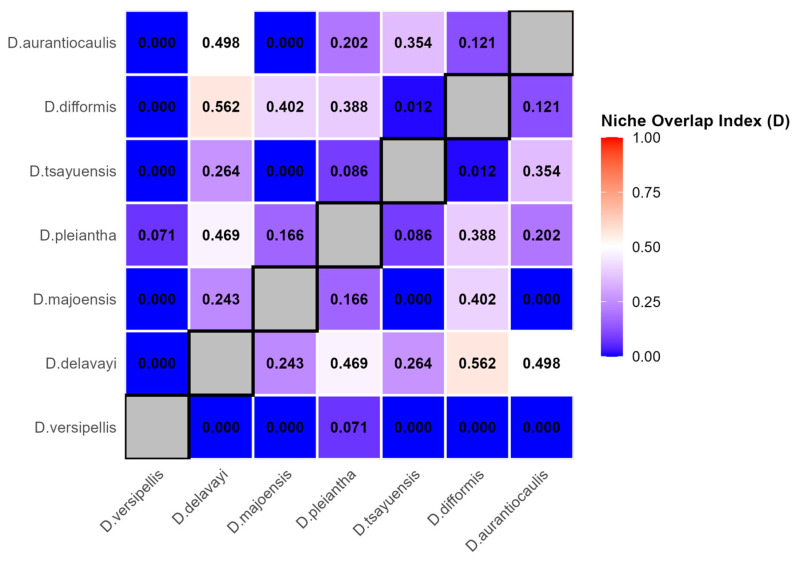
Spatial Environmental niche overlap patterns among *Dysosma* species.

**Figure 6 plants-15-00162-f006:**
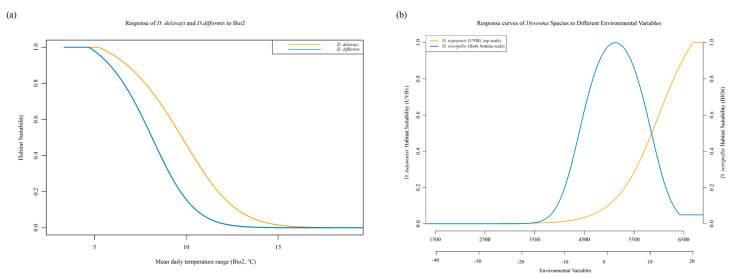
Environmental response curves for species pairs with high and low niche overlap. (**a**) High-overlap species pair; (**b**) Low-overlap species pair.

**Figure 7 plants-15-00162-f007:**
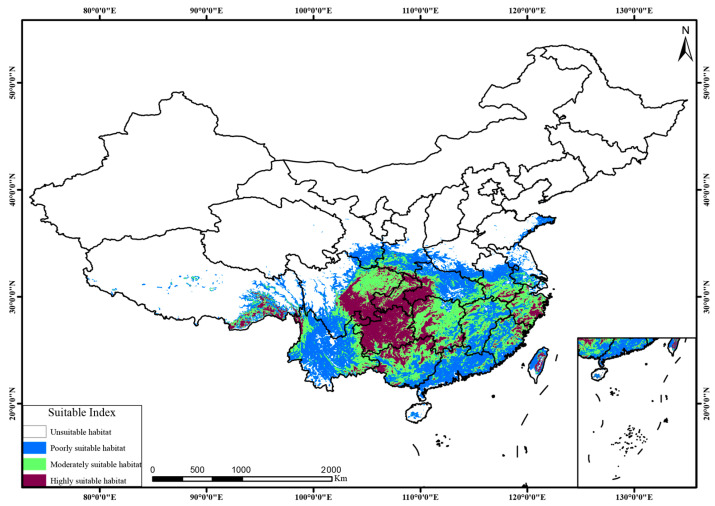
Distribution patterns of suitable habitats of *Dysosma* under the current climate scenario.

**Figure 8 plants-15-00162-f008:**
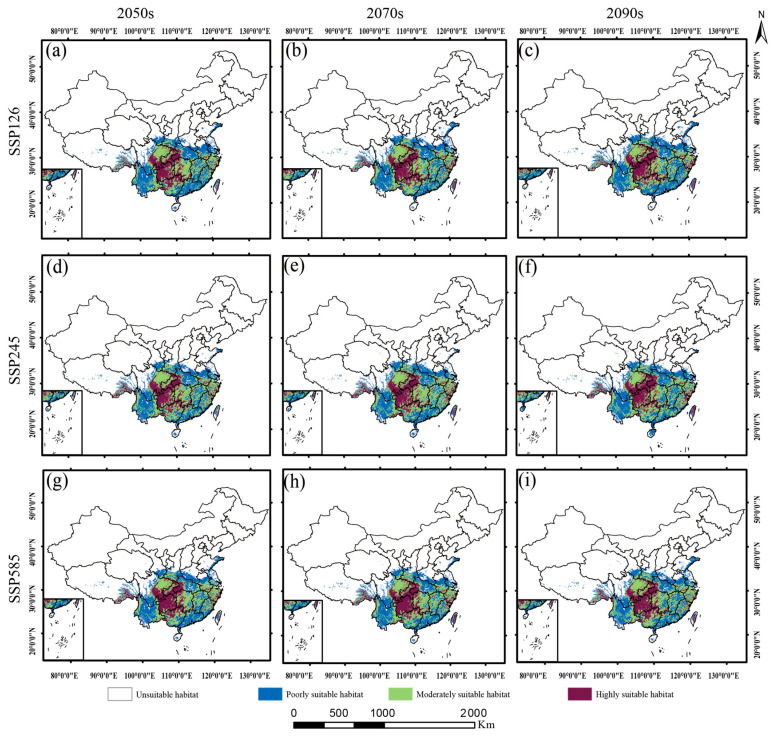
Habitat suitability of *Dysosma* under different climate scenarios: (**a**) Predicted Distribution (2050s, SSP1–2.6); (**b**) Predicted Distribution (2070s, SSP1–2.6); (**c**) Predicted Distribution (2090s, SSP1–2.6); (**d**) Predicted Distribution (2050s, SSP2–4.5); (**e**) Predicted Distribution (2070s, SSP2–4.5); (**f**) Predicted Distribution (2090s, SSP2–4.5); (**g**) Predicted Distribution (2050s, SSP5–8.5); (**h**) Predicted Distribution (2070s, SSP5–8.5); (**i**) Predicted Distribution (2090s, SSP5–8.5).

**Figure 9 plants-15-00162-f009:**
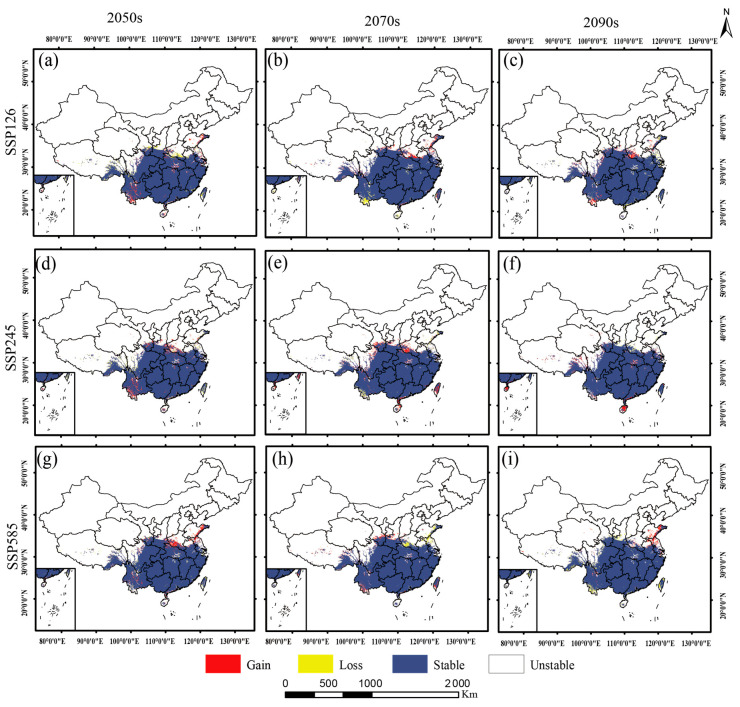
Spatial distribution patterns of *Dysosma* suitable habitat changes under future climate scenarios: (**a**) Range Dynamics (SSP1–2.6, 2050s); (**b**) Range Dynamics (SSP1–2.6, 2070s); (**c**) Range Dynamics (SSP1–2.6, 2090s); (**d**) Range Dynamics (SSP2–4.5, 2050s); (**e**) Range Dynamics (SSP2–4.5, 2070s); (**f**) Range Dynamics (SSP2–4.5, 2090s); (**g**) Range Dynamics (SSP5–8.5, 2050s); (**h**) Range Dynamics (SSP5–8.5, 2070s); (**i**) Range Dynamics (SSP5–8.5, 2090s).

**Figure 10 plants-15-00162-f010:**
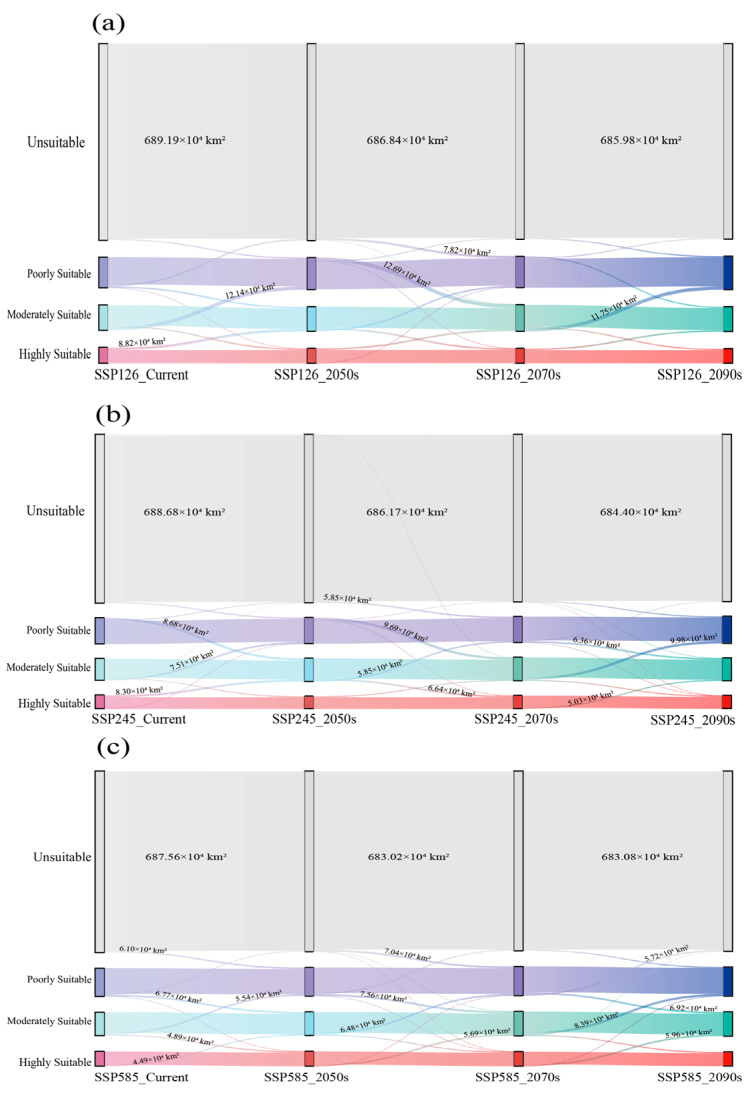
Spatiotemporal shifts in habitat suitability levels of *Dysosma* under future climate scenarios: (**a**) Spatiotemporal shifts (SSP1–2.6); (**b**) Spatiotemporal shifts (SSP2–4.5); (**c**) Spatiotemporal shifts (SSP5–8.5).

**Figure 11 plants-15-00162-f011:**
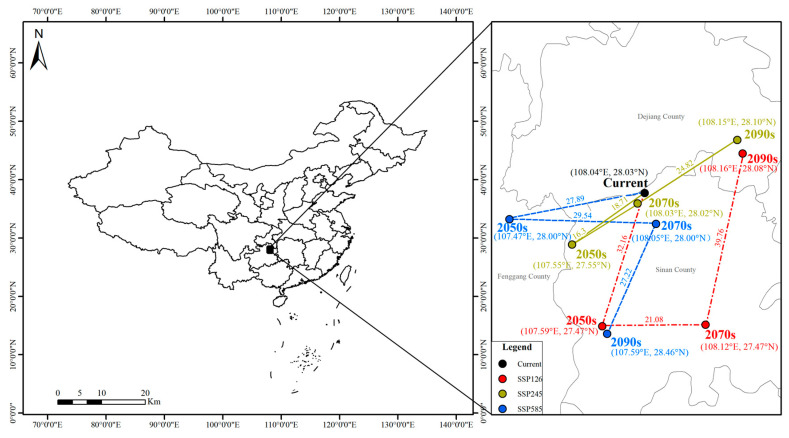
Centroid dynamics of highly suitable habitats of *Dysosma* under future climate scenarios.

**Figure 12 plants-15-00162-f012:**
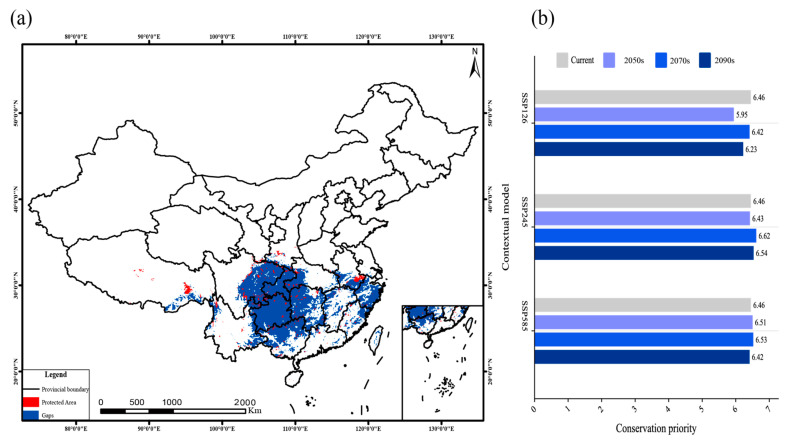
Habitat protection gaps for *Dysosma* under the current scenario: (**a**) Map of protection gaps; (**b**) Priority protection bar charts under three future scenarios.

**Figure 13 plants-15-00162-f013:**
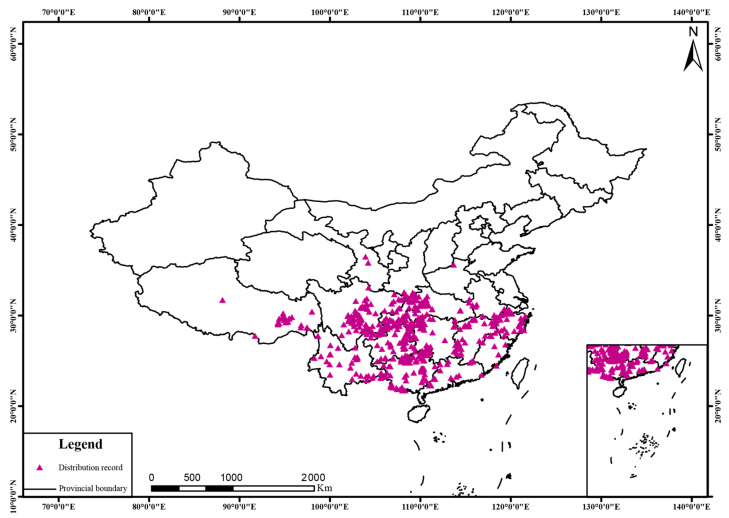
Distribution patterns of *Dysosma* species in China.

**Figure 14 plants-15-00162-f014:**
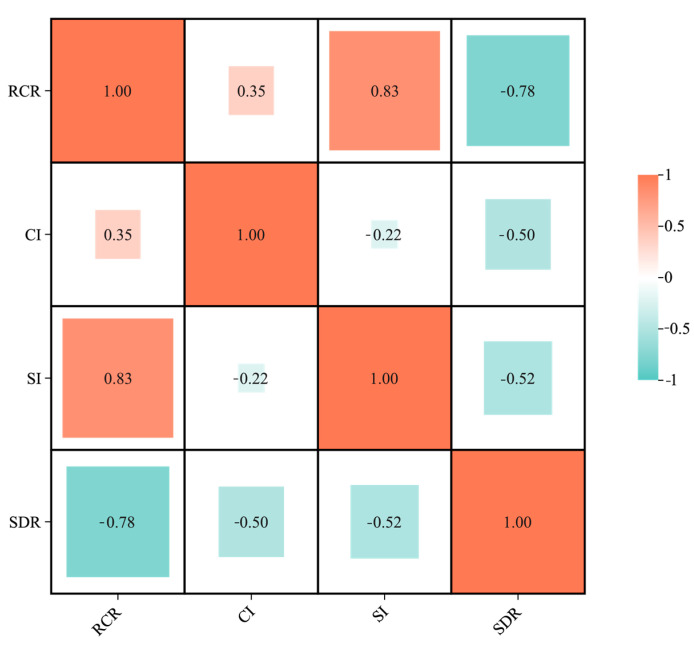
Correlation features of habitat suitability change indicators for the genus *Dysosma*.

**Table 1 plants-15-00162-t001:** Contribution (%) of key environmental drivers for species pairs with high and low niche overlap.

Species	Bio2	Bio6	Bio7	UVB1	Contributions (%) of the Top Four Environmental Variables
*D. delavayi*	13.9	-	46.2	-	84.5
*D. difformis*	46.8	39.6	-	-	91.4
*D. versipellis*	40	36.7	-	-	85.4
*D. tsayuensis*	3.3	31.1	-	52.9	95.1

**Table 2 plants-15-00162-t002:** The suitable habitat area (km^2^) of the plants of the genus *Dysosma * under different climate scenarios.

Suitability Class	Current	SSP126			SSP245			SSP585		
		2050s	2070s	2090s	2050s	2070s	2090s	2050s	2070s	2090s
HS	56.13	52.20	52.32	51.23	51.28	54.30	54.14	56.58	55.10	53.83
MS	89.49	86.42	94.06	87.57	95.50	97.02	93.64	90.40	93.39	92.9
PS	107.40	113.24	110.43	118.25	107.37	106.75	111.51	109.38	111.58	110.95
TS	253.02	251.86	256.82	257.04	254.16	258.07	259.29	256.37	260.07	257.68

**Note:** Projected habitat area (km^2^) under future climate scenarios. Suitability classes are defined as: HS, Highly Suitable; MS, Moderately Suitable; PS, Poorly Suitable; TS, Total Suitable Area.

**Table 3 plants-15-00162-t003:** Characteristics of *Dysosma* habitat suitability changes under future climate scenarios.

Scenario	Time Period	*RCR* (%)	*CI* (%)	*SI* (%)	*SDR* (%)	Dynamic Stage
SSP126	Current–2050s	−0.46	3.98	97.90	0.44	Stable Period
	2050s–2070s	1.97	4.40	98.88	0.30	
	2070s–2090s	0.09	2.97	98.66	0.48	
	Period Mean	0.53	3.78	98.48	0.41	
SSP245	Current–2050s	0.45	3.45	98.64	0.43	Slow Expansion Period
	2050s–2070s	1.54	3.43	99.19	0.30	
	2070s–2090s	0.47	2.78	98.98	0.43	
	Period Mean	0.82	3.22	98.94	0.39	
SSP585	Current–2050s	1.32	3.53	99.04	0.32	Transition Period
	2050s–2070s	1.44	4.27	98.72	0.35	
	2070s–2090s	−0.92	3.59	97.89	0.40	
	Period Mean	0.61	3.80	98.55	0.36	

Note: *RCR*, *CI*, *SI*, and *SDR* represent Relative Change Rate, Change Intensity Index, Stability Index, and Spatial Displacement Rate, respectively.

**Table 4 plants-15-00162-t004:** Significance tests of centroid migration distances of *Dysosma* under different climate scenarios.

Emission Pathway	Sample Size (*N*)	Mean Rank	Chi-Square	Degrees of Freedom (df)	*p*-Value
SSP126	3	6.67	4.36	2	0.11
SSP245	3	2.33			
SSP585	3	6			

Significant at *p* < 0.05.

**Table 5 plants-15-00162-t005:** Sources of *Dysosma* Distribution Data.

Data Source	*D. aurantiocaulis*	*D. difformis*	*D. tsayuensis*	*D. pleiantha*	*D. majoensis*	D. *delavayi*	*D. versipellis*	Total
CVH	1	54	16	105	29	89	191	485
GBIF	14	33	22	65	8	26	123	291
Survey					1	1	4	6
								782

**Table 6 plants-15-00162-t006:** Comparison of spatial thinning results for geographic distribution data at the genus and species levels.

Species	Number of Samples	
	Before Spatial Thinning	After Spatial Thinning
*D. aurantiocaulis*	15	8
*D. difformis*	87	67
*D. tsayuensis*	38	22
*D. pleiantha*	170	126
*D. majoensis*	38	32
*D. delavayi*	116	84
*D. versipellis*	318	259
Subtotal		598
Dysosma	782	534
Subtotal		534

**Table 7 plants-15-00162-t007:** MaxEnt Model Parameter Settings for *Dysosma* at the Genus and Species Levels.

Species	Number of Samples	Random Test Percentage	Feature Combination Multiplier	Regularization Multiplier
*Dysosma*	534	25	LQ	0.5
*D.versipellis*	259	30	LQ	0.5
*D.pleiantha*	126	30	LQ	2
*D.delavayi*	84	30	LQ	0.5
*D.diformis*	67	30	LQ	2.5
*D.majoensis*	32	40	LQH	3.5
*D.tsayuensis*	22	40	LQ	2.5
*D.aurantiocaulis*	8	0	L	2

**Note:** Feature types used in MaxEnt models are indicated as follows: L = Linear, Q = Quadratic, H = Hinge. Combinations such as LQ or LQH indicate that multiple feature types were selected simultaneously.

**Table 8 plants-15-00162-t008:** Indicators of predicted changes in the suitable habitat of the genus *Dysosma*.

Indicator	Index	Formula	Ecological Significance
Change Intensity Indicator	Relative Change Rate (%)	$RCR=\frac{A_{final}-A_{initial}}{A_{initial}}\times 100\%$	Quantifies the magnitude of change in the total area of suitable habitats [102].
	Change Intensity (%)	$CI=\frac{A_{expansion}+A_{contraction}}{A_{initial}}\times 100\%$	Characterizes the overall activity level of changes in suitable habitats [103].
Stability Index	Stability Index (%)	$SI=\frac{A_{stable}}{A_{initial}}\times 100\%$	Assesses the ability of core habitats to persist [104]
Spatial Pattern Indicator	Spatial Displacement Rate	$SDR=\frac{min\left(A_{expansion},A_{contraction}\right)}{A_{expansion}+A_{contraction}}$	The range of this indicator is from 0 to 0.5, with higher values indicating stronger spatial reorganization [105].

## Data Availability

Data are contained within the article.

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
