# Peer review of "Ecological Niche Differentiation and Distribution Dynamics Revealing Climate Change Responses in the Chinese Genus Dysosma"

_plants, 2026, doi:10.3390/plants15010162_

Round 1
Reviewer 1 Report
Comments and Suggestions for Authors
The manuscript by Rui Chen et al. aims to study the impacts of climate change on the distribution of “Ghostroot” or podophyllum, a group herbaceous perennial species belonging to the genus Dysosma in China. To this end, the authors model the distribution of the genus, based on 534 distribution points of 8 species, with 25 environmental variables as predictors. The authors use MaxEnt modelling algorithm to simulate the current suitable habitats of Dysosma species and predict the spatio-temporal changes of these species under three future climate scenarios (SSP126, SSP245, and SSP585). The manuscript is well structured and written in a clear, fluid language that makes it easy to follow.
Major comments:
- The authors implement accepted best practices, using a popular algorithm, along with procedures to improve the processing of information (e.g. ENM Tools) and to improve model fiiting procedure, such as ENMEval. However, regarding the core literature used to support the choice of the modelling algorithm (MaxEnt), the references used may be supported with more recent studies, to allow readers access to updated methods and practices. Please note that GARP and ENFA have been less used in recent years, being replaced by other algorithms, such as Random Forest or Boosted Regression Trees and Extreme Gradient Boosted Trees, or ensemble models such as Biomod2. In this regard, the recent work by Elith et al. (2020) and Valavi et al. (2022) may be a good option to provide readers with a current perspective.
- Regarding the taxonomic status of the genus, please note that the genus Dysosma is not universally recognised by this name, as some authorities include the plants in the genus This is the case of the World Checklist of Vascular Plants, and the Plants of the World Online resource maintained by the Kew Royal Botanical Gardens in Britain. While I understand that there may be discrepancies in the taxonomic placement of the genus, I would recommend the authors provide this information, allowing readers to be aware that other taxonomic authorities consider Dysosma as including only those Podophyllumspecies which originate in China.
- While I understand the underlying rationale is that there is urgent need to develop scientific knowledge to support management and conservation of these species, I would provide references to support the statement indicated in lines 82-85. Perhaps this sentence could be supported with additional literature, providing a general context for an international audience, thus strengthening the manuscript. One possible alternative could be Guan et al. (2010).
- Regarding the use of data to provide a genus-level model and predictions, this is an important methodological choice which should be supported by a clear logical argument and available references. Overall, Genus-level species distribution modelling can be considered adequate when addressing research questions at a broad geographic scale or at a lineage-level taxonomic description. A second situation where this approach may be considered adequate is when genera are species-poor, and the niches of the species are conservative and similar among all species in the focus genus. This could be the case in this study, but the authors should present evidence and arguments to support this choice. Please note that this approach may not be valid if the species within the genus are known to be ecologically differentiated, showing strong elevational, habitat, or climatic segregation. As a result, I suggest the authors consider discussing these assumptions and limitations in both the methods and discussion sections.
- Regarding the variation of geographic distribution, I found it particularly interesting that under the three scenarios examined (SSP126, SSP245 and SSP585) the predicted trajectories of centroids of high suitability areas for species in the genus Dysosma, differ strongly between scenarios, as shown in Figure 9. However, when examining the habitat suitability and the spatial distribution of the suitable habitat areas for species in the genus Dysosma species under future climate scenario models, shown in Figures 6 and 7, these seem to be quite stable, a fact that is also evident in Figure 8. This seems a bit counterintuitive and may reflect the changes in the less suitable habitats. I would encourage the authors to discuss this aspect.
Minor comments:
Lines 50-51: I would suggest you provide some specific examples and references to support this sentence.
Lines 51-54: Please revise this sentence, which seems to be too long and awkward. I would suggest a phrase such as: “Given the increasing impacts of climate change and extreme weather events, understanding plant species distribution patterns and predicting future changes is a critical component of biodiversity protection, along with both in-situ and ex-situ conservation.”
Line 76. I would recommend adding a citation for the Chinese classic text "Shennong Ben Cao Jing", a translation of it or an article that introduces it. This would allow readers to become familiar with its relevance and importance.
Suggested References:
Elith J, Graham C, Valavi R, Abegg M, Bruce C, Ferrier S, Ford A, Guisan A, Hijmans RJ, Huettmann F, Lohmann L. Presence-only and presence-absence data for comparing species distribution modeling methods. Biodiversity informatics. 2020 Jul 22;15(2):69-80.
Guan, B. C., Fu, C. X., Qiu, Y. X., Zhou, S. L., & Comes, H. P. (2010). Genetic structure and breeding system of a rare understory herb, Dysosma versipellis (Berberidaceae), from temperate deciduous forests in China. American Journal of Botany, 97(1), 111-122.
Valavi R, Guillera‐Arroita G, Lahoz‐Monfort JJ, Elith J. Predictive performance of presence‐only species distribution models: a benchmark study with reproducible code. Ecological monographs. 2022 Feb;92(1):e01486.
Author Response
Answer: This revision includes the validation of the accuracy of the genus-level model predictions and the calculation of the niche overlap index based on 25 environmental variables. The references you provided have also been cited, and the specific classification system used is clearly stated in the text. Regarding the anomalous phenomena observed in the centroid analysis, we have provided a detailed explanation in the manuscript.
Reviewer 2 Report
Comments and Suggestions for Authors
This manuscript presents a modeling study using the MaxEnt species distribution model to predict current and future suitable habitats for the genus Dysosma, an endangered group of plants endemic to China. The authors analyze 534 distribution points and 25 environmental variables to simulate habitat suitability under current conditions and three Shared Socioeconomic Pathways (SSP126, SSP245, SSP585) for the periods 2050, 2070, and 2090. Key findings include the identification of temperature-related variables (e.g., Bio6 and Bio2) as dominant drivers, stable core habitats in southern China (particularly Guizhou), and varying spatiotemporal dynamics under different scenarios, with centroid migrations remaining within Guizhou Province.
The study is timely and relevant, given the conservation status of Dysosma species and the impacts of climate change on biodiversity. It contributes to the literature on species distribution modeling for rare plants and provides practical implications for conservation. The use of MaxEnt is appropriate, and the inclusion of dynamic indices (e.g., relative change rate, stability index) adds value by quantifying spatiotemporal shifts. Figures and tables are generally clear and supportive, and the manuscript is well-structured.
However, the manuscript has several deficiencies that limit its rigor and depth. These include methodological ambiguities (e.g., lack of detail on data sourcing, variable selection, and model assumptions), inconsistencies in data presentation (e.g., minor discrepancies in habitat areas across text and tables), and superficial discussion of limitations, ecological implications, and conservation strategies. The genus-level approach treats all eight species as ecologically uniform, which may overlook interspecific differences. Additionally, the discussion section (based on the provided pages) is underdeveloped, focusing primarily on summarizing results without deeper integration with broader ecological theory, climate refugia concepts, or policy recommendations. The authors should significantly deepen the discussion to explore these aspects, such as linking findings to evolutionary adaptations, potential biotic interactions, or specific in-situ/ex-situ conservation actions.
Deficiencies and Weaknesses
- Methodological Gaps: The manuscript lacks detailed information on the sourcing and filtering of the 534 distribution points (e.g., from herbaria, field surveys, or databases like GBIF). Potential biases, such as sampling effort or georeferencing errors, are not addressed. The selection of 25 environmental variables is mentioned but not listed explicitly, and there is no mention of correlation analysis to mitigate multicollinearity, which is a common issue in MaxEnt modeling. The choice of model parameters (regularization multiplier = 0.5, feature class = LQ) is stated but not justified—why not default settings or optimization via ENMeval?
- Genus-Level vs. Species-Level Modeling: Treating the entire genus as a single unit assumes ecological similarity among the eight species, but Dysosma species may have distinct niches (e.g., altitudinal preferences or habitat specificities). This could lead to overgeneralized predictions. No sensitivity analysis is provided to compare genus-level vs. species-specific models.
- Climate Data and Scenarios: The future climate data source (e.g., WorldClim, CMIP6) is implied but not specified, including which General Circulation Models (GCMs) were used or if an ensemble approach was applied. This reduces reproducibility. Soil variables (e.g., pH, texture) are included in figures but downplayed in text, despite potential importance for rhizomatous plants like Dysosma.
- Data Inconsistencies and Typos: There are minor discrepancies, e.g., current high-suitability area is listed as 56.13 × 10⁴ km² in Table 1 but referenced differently in text (e.g., 56.58 for SSP585-2050). Author affiliations and emails have redundancies (e.g., Mao Li listed twice). Units in tables (km²) are clear, but some figures (e.g., Fig. 1) have labels like "omission rate" without full explanation in captions.
- Analytical Limitations: The dynamic indices (e.g., relative change rate, stability index, spatial displacement rate) are useful but their formulas or calculation methods are not provided in the methods section (assuming it's in the remaining pages, but not visible here). Centroid analysis is descriptive but lacks statistical testing (e.g., for significant shifts). The manuscript acknowledges MaxEnt limitations (e.g., no biotic interactions) but does not discuss how these affect results.
- Underdeveloped Discussion: The discussion is brief and repetitive of results, lacking depth on broader implications. For instance, why is Guizhou a refuge—due to topography, microclimates, or other factors? How do findings align with other studies on Berberidaceae or similar genera? Conservation gaps are mentioned in objectives but not elaborated (e.g., overlap with protected areas). No discussion of uncertainties like dispersal limitations or human impacts (e.g., habitat fragmentation).
- Presentation and Clarity: Some sections are wordy (e.g., introduction repeats climate change impacts). English language could be polished for grammar and flow (e.g., "Ghostroot" should be consistently capitalized if a common name). References are appropriate but could include more recent MaxEnt applications (post-2020).
Questions for the Authors
To address the deficiencies and deepen the discussion, please respond to the following 8 questions in your revision:
- How were the 534 distribution points obtained and filtered? Please provide details on sources (e.g., GBIF, CVH), any removal of duplicates or outliers, and potential sampling biases.
- Why was a genus-level model used instead of species-specific models? Did you conduct a sensitivity analysis to check for interspecific differences in environmental responses?
- Which climate data source and GCM(s) were used for future projections? If multiple GCMs, why not an ensemble mean to reduce uncertainty?
- How were the 25 environmental variables selected, and was multicollinearity addressed (e.g., via VIF or correlation matrices)? Please list all variables in a supplementary table.
- What are the exact formulas for the dynamic indices (e.g., relative change rate, stability index, spatial displacement rate)? How do these indices account for habitat fragmentation?
- The centroid remains in Guizhou—could you deepen the discussion on why this region acts as a climatic refuge, perhaps linking to topographic heterogeneity or paleoclimate history?
- How do soil variables (mentioned in figures) interact with climatic factors? Please expand the discussion on their role, given Dysosma's rhizomatous growth.
- What are the specific conservation gaps identified (e.g., overlap with nature reserves)? Deepen the discussion on practical strategies, such as ex-situ cultivation or assisted migration, under the SSP585 contraction scenario.
This manuscript has a solid foundation but requires major revisions for clarity, rigor, and depth. Particularly, expand the discussion section to at least double its current length, integrating ecological theory and conservation applications. Provide supplementary materials for reproducibility (e.g., variable list, R/ArcGIS code snippets). I look forward to seeing the revised version.
Author Response
- How were the 534 distribution points obtained and filtered? Please provide details on sources (e.g., GBIF, CVH), any removal of duplicates or outliers, and potential sampling biases.”
Answer: The source information for the 534 distribution points has been added in Table 5 on lines 644-645, which provides detailed sources.。
- “Why was a genus-level model used instead of species-specific models? Did you conduct a sensitivity analysis to check for interspecific differences in environmental responses?
Answer: To avoid overgeneralizing the predictions, we have added a validation of the genus-level model predictions. The specific method can be found in Section 4.5 (lines 738-772). Interestingly, we found that Dysosma tsayuensis is an ecologically specialized species, and its distribution is highly dependent on the annual average ultraviolet B radiation (Uvb1, contribution rate 52.9%).。
3.Which climate data source and GCM(s) were used for future projections? If multiple GCMs, why not an ensemble mean to reduce uncertainty?”
Answer: The corresponding future climate data sources are described specifically in lines 706-709, where we explain that the GCMs were used. Regarding soil variable data, we have discussed this in lines 605-608.’
4.How were the 25 environmental variables selected, and was multicollinearity addressed (e.g., via VIF or correlation matrices)? Please list all variables in a supplementary table.
Response: The correlation analysis for the 25 environmental variables is presented in lines 667-668, with the corresponding correlation heatmap provided in Figure S5. The specific parameters for model tuning are located in lines 736-737.
5.What are the exact formulas for the dynamic indices (e.g., relative change rate, stability index, spatial displacement rate)? How do these indices account for habitat fragmentation?
The formula for the dynamic index can be found in lines 794-795, with the specific formula added in Table 8. The centroid analysis has been added in lines 806-810, where the Kruskal-Wallis test was used. In lines 609-613, we provide a detailed discussion on biotic interactions.
- The centroid remains in Guizhou—could you deepen the discussion on why this region acts as a climatic refuge, perhaps linking to topographic heterogeneity or paleoclimate history?
The discussion has already been specifically addressed in the article
7.What are the specific conservation gaps identified (e.g., overlap with nature reserves)? Deepen the discussion on practical strategies, such as ex-situ cultivation or assisted migration, under the SSP585 contraction scenario.
The discussion has already been specifically addressed in the article
Round 2
Reviewer 1 Report
Comments and Suggestions for Authors
I have read the revised manuscript by Rui Chen et al., which aims to study the impacts of climate change on the distribution of “Ghostroot”, a group herbaceous perennial species belonging to the genus Dysosma in China. The manuscript is well structured and written in a clear, fluid language that makes it easy to follow. The authors have addressed the concerns I pointed out in the previous version, namely:
- Suggested references. The authors have referenced recent articles, allowing them to present readers with updated methods and practices, as well as providing a supporting reference for their mention of the classic text Shennong Bencao Jing.
- The authors have clarified their use of the Flora of China (FOC) taxonomic system.
- The authors have addressed the concern regarding the use of data to provide a genus-level distribution model and predictions. The manuscript now complements the genus-level modelling with an analysis of niche differentiation among species, providing a stronger, more robust analysis.
Given these changes, I have no further concerns regarding this manuscript.
Minor comments:
lines 694 to 695 and 718 and Table 6: please use italics for the scientific names. I suggest the manuscript be thoroughly proofread for similar instances.
Reviewer 2 Report
Comments and Suggestions for Authors
The authors have thoroughly revised the manuscript entitled “Prediction of Suitable Habitats and Spatiotemporal Variation Characteristics of the Genus Dysosma (Berberidaceae) Based on the MaxEnt Model.” The revised version shows substantial improvement in both content and presentation. The authors have carefully and satisfactorily addressed the reviewers’ comments, with notable enhancements to the clarity of the methodology, the quality and interpretability of figures and tables, and the overall coherence of the text. The analyses are now more transparent and robust, and the discussion better contextualizes the results within existing literature, highlighting the relevance of the findings for biogeography and conservation planning of Dysosma. In its current form, the manuscript meets the standards of the journal, and I therefore recommend acceptance for publication.